# Ro60—Roles in RNA Processing, Inflammation, and Rheumatic Autoimmune Diseases

**DOI:** 10.3390/ijms25147705

**Published:** 2024-07-14

**Authors:** Ranjeet Singh Mahla, Esther L. Jones, Lynn B. Dustin

**Affiliations:** The Kennedy Institute of Rheumatology, University of Oxford, Oxford OX3 7FY, UK; esther.jones@univ.ox.ac.uk

**Keywords:** Ro60, inflammation, Sjögren’s disease, RNA editing, autoantibodies, autoimmunity

## Abstract

The Ro60/SSA2 autoantigen is an RNA-binding protein and a core component of nucleocytoplasmic ribonucleoprotein (RNP) complexes. Ro60 is essential in RNA metabolism, cell stress response pathways, and cellular homeostasis. It stabilises and mediates the quality control and cellular distribution of small RNAs, including YRNAs (for the ‘y’ in ‘cytoplasmic’), retroelement transcripts, and misfolded RNAs. Ro60 transcriptional dysregulation or loss of function can result in the generation and release of RNA fragments from YRNAs and other small RNAs. Small RNA fragments can instigate an inflammatory cascade through endosomal toll-like receptors (TLRs) and cytoplasmic RNA sensors, which typically sense pathogen-associated molecular patterns, and mount the first line of defence against invading pathogens. However, the recognition of host-originating RNA moieties from Ro60 RNP complexes can activate inflammatory response pathways and compromise self-tolerance. Autoreactive B cells may produce antibodies targeting extracellular Ro60 RNP complexes. Ro60 autoantibodies serve as diagnostic markers for various autoimmune diseases, including Sjögren’s disease (SjD) and systemic lupus erythematosus (SLE), and they may also act as predictive markers for anti-drug antibody responses among rheumatic patients. Understanding Ro60’s structure, function, and role in self-tolerance can enhance our understanding of the underlying molecular mechanisms of autoimmune conditions.

## 1. Introduction

Ro60 (also known as Sjögren’s disease (SjD) autoantigen A2 (SSA2), RO60, or TROVE2) is an RNA-binding protein (RBP). Approximately 300 human RNA-binding proteins are known; however, only a few are characterised for their structures and functions. Ro60/SSA2 is essential for disease and developmental processes [1,2]. Its distribution within cells varies between the nucleus, nucleolus, and cytoplasm. It is directly involved in maintaining cellular homeostasis and mediating RNA metabolism, stress responses and DNA replication. Although Ro60 is ubiquitously expressed, its expression can diminish with age [1]. Ro60 is a core component of nucleocytoplasmic RNP complexes. Many RNA molecules such as YRNAs, Alu retroelement transcripts, LINE1 retroelement transcripts, and 5S rRNAs directly bind with Ro60. Ro60/SSA2 interactions regulate stability and cellular distribution and serve as quality control for these RNA molecules [3,4,5]. Ro60’s functions intersect with those of other RNA-binding proteins and the complex pathways that regulate cellular RNA transcription, splicing, stability, localisation, translation, and disposal throughout the cell’s lifespan [6,7,8,9,10,11]. The protein part of the RNP complex, Ro60/SSA2, has a molecular weight of 60 kDa and thus was assigned the name ‘Ro60’. The combined weight of the Ro60 RNP complex varies with bound RNA moieties and interacting protein partners [12]. In Hela S100 cells extracts, the molecular weight of the Ro60 RNP complex was estimated to vary from 150 kDa to 550 kDa [1,12]. Such variations in molecular weight could be attributed to compositional differences. The Ro60 RNP complex is not merely a combination of Ro60 and YRNAs but also involves other small RNA moieties and interacting proteins, including the antigenic protein La/SSB [1,12,13,14].

Unlike sequence-specific RBPs, Ro60 scavenges small RNA molecules that fail to bind their cognate RBPs, thereby contributing to cellular homeostasis. Ro60/SSA2 is evolutionarily conserved and found in all animal species, with orthologues also reported in some bacterial species [15]. This review highlights how Ro60 mediates small RNA quality control and how interactions between Ro60 and small RNA can lead to the immunogenic exposure of Ro60 RNP complexes. Such exposure may drive inflammatory responses and autoantibody production.

## 2. Structure of Ro60

Progress has been made in understanding the domain architecture of Ro60 by examining its counterparts in the African clawed frog (*Xenopus laevis*) [16,17] and the extremophilic bacterium *Deinococcus radiodurans* [15]. *X. laevis* Ro60, which shares 78% identity with human Ro60, comprises two distinct domains: the vWFA (von Willebrand factor A) domain and the TROVE (telomerase, Ro, vault) domain (Figure 1A) [17]. The vWFA domain is present in many proteins, including those that mediate cell adhesion, migration, homeostasis, signal transduction, and immune defence mechanisms [18]. In Ro60, the vWFA domain stabilises the central cavity of the TROVE domain [17]. The vWFA domain contains a metal ion-dependent adhesion site (MIDAS; DxSxS…T…D), which binds divalent cations and mediates ligand binding and structural rearrangements. In integrins and other cell adhesion proteins, the vWFA domain functions as a ligand binding site [19]. The vWFA domain also interacts with other domains such as the NACHT-NTPase domain, telomerase-associated protein 1 (TEP1) N-terminal domain, and WD-40 repeats [20]. The TROVE domain comprises seven α-helical repeats, arranged in a circular format with a central toroid hole of 10Å–15Å. This toroid hole can accommodate the binding of many small single-stranded RNAs (ssRNAs) and misfolded RNAs [17]. The TROVE domain comprises ~300–500 residues and is present in TEP1 (also known as TROVE1) and Ro60. The central TROVE cavity is composed of positively charged amino acid residues that enable it to interact with negatively charged RNA molecules, particularly ssRNAs [20].

## 3. Ro60 in Quality Control and Cellular Distribution of Small RNAs

### 3.1. YRNAs

Ro60 is crucial for the quality control, cellular distribution, and structural and functional surveillance of small RNAs, including YRNAs and rRNAs. YRNAs are small (84–113 nucleotides (ntd) long) noncoding RNA molecules. In humans, there are four YRNAs: human RNA Ro60 associated Y1 (hRNY1/hY1; 113 ntd; NR_004391.1), hRNY3/hY3 (102 ntd; NR_004392.1), hRNY4/hY4 (96 ntd; NR_004393.1), and hRNY5/hY5 (85 ntd; NR_001571.2). Genes encoding human YRNAs (hY1, hY3, hY4, hY5) are located on chromosome 7q36.1. The letter Y refers to ‘cYtoplasmic’, although these RNAs are also present in the nucleus [22]. YRNAs possess characteristic stem-loop structures with two main stems and single-stranded bulges. Ro60 and La/SSB (another SjD autoantigen and RBP) play essential roles in YRNA stabilisation, nuclear export, and cellular distribution.

Ro60 and La binding sites are conserved among all four YRNAs. YRNAs are stem-bulge RNAs, with lower stems and upper stems (Figure 1B). In the nucleus, YRNAs’ lower stem loops stably bind with RBPs, while upper stem-loop regions mediate the regulation of DNA replication [22,23]. La binds to YRNA’s 3′ UCUUUU sequences, whilst Ro60 interacts with YRNA’s evolutionarily conserved lower stem (also known as the 5′-3′ stem) region [24]. The interaction between Ro60 and YRNAs relies on specific conserved ntd sequences, indicating a sequence-specific binding mechanism. The ninth ntd within the stem bulge at the 5′-3′ terminus, within the conserved sequence ‘GGUC***C***GA’, is of critical importance (Figure 1C) [25,26,27]. The binding of Ro60 is essential for YRNA transport and quality control (Figure 1D). YRNAs bind to the outer surface of the toroid ring, while single-stranded RNA ntd oligomers bind to the inside of the toroid hole [17]. The Ro60 and La/SSB RNP complex exhibits considerable stability, with both Ro60 and La RBPs remaining in proximity due to protein interactions or the RBP-mediated stabilisation of bound YRNAs. This interaction helps retain these two RBPs in the nucleus [14]. Ro60-bound YRNA is exported to the cytoplasm in a process mediated by Ran GTPase [21] (Figure 1D).

Mature mRNA has 3′ poly-A tail, while mature YRNAs have a poly-U tail. Polyadenylation marks YRNAs for exonuclease degradation. However, Ro60 protects YRNAs from polyadenylation [28]. Importantly, YRNA binding to Ro60 facilitates the binding of other small RNA molecules, including 5S rRNA. These 5S rRNA form complexes with Ro60 [5,29], La/SSB [29], L5, µL5, L11, µL18, and TFIIIA [5]. Ro60 binding with 5S rRNA is prominent in the nucleoli, and such binding may be mediated by either protein–RNA or RNA–RNA interactions. The RNA-to-RNA interactions between the Ro60 RNP complex and 5S rRNA in the nucleus may be facilitated by physical interactions between hY4/hY5 and Ro60 [5]. Ribosomal protein L5, which forms the L5-rRNA RNP complex, also binds to hY5, suggesting a possible association between 5S rRNA and YRNAs through L5-RNP/Ro60-RNP interactions. L5 plays a crucial role in the cellular trafficking of 5S rRNA and serves as a core component of the ribosomal machinery. YRNAs can physically interact with ribosome-unoccupied L5-5S rRNA RNP, thereby modulating its recruitment to Ro60. Human YRNAs vary in their internal loop region sequences and sizes (Figure 1B). The 5S rRNA interactions with Ro60 largely depend on the sequences and sizes of YRNAs’ internal loop regions.

### 3.2. 5S rRNA

YRNA5, with its smaller internal loop region, is capable of binding with Ro60 and 5S rRNA, while larger internal loop regions create steric hindrance [30]. The 5S rRNA is a fundamental component of the large ribosome subunit across all three domains of life: bacteria, archaea, and eukaryotes [31]. All 5S rRNA molecules adopt a conserved secondary structure consisting of five double-stranded regions (numbered I to V) and five loops (lettered A to E), including two hairpin loops, two internal loops, and a three-helix junction hinge region (Figure 1E). In *X. laevis*, Ro60, in conjunction with the La/SSB protein, specifically binds to a variant of pre-5S rRNA that has ntd changes due to readthrough across the first termination signal, leading to 8-10 ntd extensions at the 3′ end. The presence of Ro60 and La/SSB is essential for targeting these variant pre-5S rRNAs to discard pathways, which maintain quality control of the structural and functional integrity of 5S rRNA moieties [4]. The misfolded 5S rRNA precursor adopts an alternative helical structure, which allows it to bind to the central cavity of Ro60 via a single-stranded 3′ extension [17]. Similar to the misfolded defective pre-5S rRNA, wild-type 5S rRNA can also adopt an alternative helical structure [29]. A loss of Ro60 function in macrophages and lymphocytes can result in the misprocessing of 18S and 28S rRNAs, causing alterations in RNA length and promoting the aberrant polyadenylation of these noncoding RNAs [28]. The misprocessing of rRNAs can disrupt the protein biosynthesis machinery, as rRNAs play a crucial role in protein synthesis.

### 3.3. Alu and L1 Retroelement Transcripts

Up to half of the human genome is derived from retroelements, which share sequence, transcription, and amplification characteristics with retroviruses; these include long interspersed nuclear elements (LINEs), short interspersed nuclear elements (SINEs), and endogenous retroviruses (ERVs). These elements undergo amplification through reverse transcription. Ro60 binds retroelement transcripts, and such binding reduces the ability of those transcripts to stimulate inflammation [3]. Ro60 mounts qualitative and quantitative checks on retroelement transcription and amplification. Alu is the most common retroelement in primates [32], and a small proportion of Alu elements are transcribed by RNA polymerase III. Alu transcripts have significant impacts on various cellular processes, including mRNA polyadenylation, RNA splicing, and adenosine deaminase acting on the RNA (ADAR)-mediated A-to-I editing of RNA. Ro60 knockout cell lines exhibit elevated Alu transcripts [3]. In SLE patients, the level of Ro60 autoantibodies in the circulation correlated with the levels of circulating Alu RNA and expression of an IFN-stimulated gene signature [3].

## 4. Ro60’s Role in RNA Editing

RNA editing is a post-transcriptional modification process [33] impacting protein-coding mRNA sequences, microRNA sequences, microRNA target sites, and nucleus-to-cytoplasm RNA export. Adenosine-to-inosine (A-to-I) modification is the most common RNA editing process, catalysed by ADARs [34]; A-to-I modifications modulate developmental processes and disease progression, including autoimmune diseases [35,36]. ADAR enzymes (ADAR1, ADAR2, and ADAR3) are present in nearly in all tissues but particularly abundant in the central nervous system. Ro60 is one of several proteins that can modulate the efficiency and specificity of A-to-I RNA editing by ADARs [37] (Figure 2A). ADAR1-eCLIP analysis showed that Ro60 binds RNAs close to ADAR1 editing sites and modulates RNA editing positively and negatively. The dysregulation of RNA editing is observed in autoimmune diseases, including rheumatoid arthritis (RA) and SLE. In SLE, loss of Ro60 function is associated with increased ADAR1p150, a specific isoform of ADAR1 [38]; elevated ADAR1p150 expression drives RNA editing (Figure 2B), potentially contributing to SLE pathogenesis [38]. In RA, the increased expression of ADAR1 and ADAR1p150 is associated with the increased A-to-I modification of Alu retroelement transcripts [39] (Figure 2B). Ro60 RNP complexes also interact with other RNA editing enzymes such as apolipoprotein-B mRNA editing enzyme catalytic polypeptide-like (APOBECs) [40]. APOBECs (including 3A, 3B, 3C, 3D, 3F, 3G, 3H, and AID) contribute to host defence by C to U conversions, which inhibit viral genomic integration [41]. The association of APOBECs with Ro60 RNP suggests potential interplays between RNA editing and antiviral mechanisms. APOBEC3A, APOBEC3G, and AID expression were increased in salivary gland (SG) biopsies of SjD patients compared to that in controls without SjD. APOBEC3A expression was elevated in the kidneys and blood cells of SLE patients [38,42]. Excessive RNA editing can be mutagenic. In SjD, higher expression of AID and APOBEC3G may contribute to an increased mutagenic load [42]. In summary, the Ro60-RNP complex mediates the modulation of A-to-I editing, and the dysregulation of Ro60 functions is associated with SjD, SLE, and RA pathogenesis.

## 5. Ro60 RNP Complex and Its Extracellular Presentation

The targeting of Ro60 RNPs by autoantibodies raises the question of whether and how Ro60 is exposed to the extracellular environment, thereby instigating an antibody response.

### 5.1. Components of the Ro60 RNP Autoantigen Complex

There is sparse evidence on the composition and structure of the Ro60 RNP autoantibody complex. In autoimmune diseases such as SjD, SLE, RA, and systemic sclerosis (SSc), Ro60 autoantibodies often co-exist with autoantibodies targeting Ro52 (also called TRIM21) and La/SSB [43]. The presence of these autoantibodies together might suggest the enrichment of Ro52 and La/SSB within autoantigenic RNP complexes. However, it is worth noting that Ro52 and Ro60 are distinct in their structure and function. Ro52 is an E3 ubiquitin ligase [44], whilst Ro60 is an RNA-binding protein. It seems that there is no direct interaction between Ro52 and Ro60 autoantigens, and the historical artefacts and misunderstanding can be corrected by separating out these two autoantigens from the umbrella acronym ‘SSA’. The ‘Ro’ in Ro60 and Ro52 is another historical artefact referring to a patient identifier [45]. One possibility is that calcium-dependent protein–protein bridging might play a role in bringing Ro52 into an autoimmune RNP complex [46]. Ro60 RNP complexes are preloaded with a diverse range of small RNA moieties. Juste-Dolz et al. [46] reported that autoantibody-bound Ro60 undergoes a conformational change such that antibody binding exposes a cryptic ‘Fc receptor’ within Ro60’s tertiary structure. Bound antibodies can then bridge additional Ro60 molecules to trigger immune complex (IC) growth. Antibodies bound to Ro60 may also bind via their Fc domain to Ro52’s PRY-SPRY domain [44], possibly leading to the artefactual detection of both autoantigens in Ro60 immunoprecipitants. Ro60 is a calcium (Ca^2+^)-binding protein, and the presence of Ca^2+^ ions is essential for protein–protein interactions between the antigenic protein La and Ro60 [47], as well as the enhancement of Ro60 antigenicity. The Ca^2+^ ions are also vital for maintaining the native conformation of the Ro60 protein [48], enabling its binding to other antigenic proteins and anti-Ro60 antibodies. Furthermore, chemical modifications of Ro60 can enhance its antigenicity. The immunisation of mice with 4-hydroxy-2-nonenal (HNE)-modified Ro60 results in a breach of self-tolerance and the onset of SjD-like phenotypes in mice [49]. Therefore, the Ro60-RNP complex should be studied for post-transcriptional and post-translational modifications, including phosphorylation, ubiquitination, sumoylation, citrullination, methylation, and glycosylation.

### 5.2. Ro60 RNP Complex Transportation and Delivery to Antigen-Presenting Cells (APCs)

The Ro60-RNP complex is intracellular, and the mechanisms for its delivery to APCs are poorly understood. Based on available evidence, we propose three possible mechanisms through which the Ro60 RNP complex may be exposed in the extracellular space: small RNA-mediated extracellular translocation, extracellular vesicle (EV)-mediated delivery of Ro60 RNPs to APCs, or apoptotic and pyroptotic bleb-mediated exposure and delivery to APCs (Figure 3A). Recent evidence suggests YRNAs are crucial for the XPO5/Ran GTPase-mediated transport of Ro60 from the nucleus to the cytoplasm and for its subcellular localisation [50] (Figure 1D). In mouse fibroblasts, Ro60 RNPs are translocated to the cell surface during early apoptosis in a manner dependent on their binding of mouse YRNA3 (mYR3) [50]. An Ro60 mutant lacking mYR3 binding (H187S) was unable to translocate to the plasma membrane. Similarly, the knockdown of mYR3 resulted in the nuclear accumulation of Ro60 [50]. However, an Ro60 mutant that retained mYR3 binding but lacked binding to 5S rRNA was still able to translocate to the plasma membrane [50]. This suggests that both Ro60 and its bound small RNAs influence its localisation.

All cells synthesise and secrete EVs for short- and long-distance cell-to-cell communication; these EVs can deliver biomolecules, including nucleic acids, lipids, and proteins [51]. EVs may serve as sources of autoantigens, contributing to a breach of self-tolerance. For example, EVs (exosomes) derived from the epithelial cells of SjD patients were enriched in autoantigens such as La/SSB, Ro/SSA, and Sm RNPs [52] (Figure 3A). APCs can capture EVs by phagocytosis or receptor-mediated endocytosis (Figure 3B), process the antigens, and present them to T lymphocytes, thereby initiating an immune response [52]. EVs carry Ro60-binding small RNAs and small RNA-derived fragments [53,54]. Their YRNA content can vary across source cells, as shown for PBMCs vs. neutrophils [53]. EVs derived from murine cardiosphere cells were found to be enriched in YRNA4 fragments [55], while EVs derived from cell lines were rich in YRNA5 fragments [56]. The enrichment of specific autoantigens and small RNA molecules in EVs may contribute to breaching self-tolerance in autoimmune diseases.

Cell death may also expose Ro60 RNP complexes to the extracellular space. Apoptosis can occur in response to a wide range of stimuli, such as DNA damage, growth factor deficiency, and viral infection, and is considered non-immunogenic. However, immunogenic apoptosis triggers are less well understood. During apoptosis, the subcellular redistribution of specific autoantigens, including Ro60, Ro52, and La/SSB antigens, has been observed [57]. Autoantigenic targets associated with SLE are packaged into two types of blebs, originating from the fragmented nucleus and endoplasmic reticulum, in response to UV damage [58]. The extracellular exposure of cryptic nuclear neoantigen epitopes, such as Ro60 and La/SSB, during apoptosis can lead to immune activation [57,59]. The exposed neoantigen epitopes from nuclear antigens are referred to as apotopes [60]. The mechanism by which these epitopes become exposed in the extracellular environment can be investigated in vitro using stimulants that promote apoptosis, such as tumour necrosis factor-alpha (TNFα). The induction of apoptosis can also result in the cleavage of nuclear antigens [61], potentially exposing neoepitopes. Pyroptotic and necroptotic cell death are pro-inflammatory lytic mechanisms that may release cytosolic contents.

## 6. Innate Immune Sensing of Ro60 RNP Complexes and Inflammation

APCs can engulf Ro60 RNP complexes with bound small RNA moieties. These small RNA molecules can engage with pattern recognition receptors, serving as ‘danger’ signals to drive inflammation and adaptive immune responses (Figure 3B). APCs can internalise Ro60-IC via Fc and complement receptors. Following uptake, RNAs derived from the Ro60 RNP complex might activate endosomal TLRs and cytoplasmic RNA sensors (Figure 3B). The outcomes of innate immune activation might depend on types of small RNA contained within the Ro60 RNP complex; some small RNAs may promote pro-inflammatory responses [62], potentially modulating immune responses. In this section, we emphasise the roles of Ro60 RNP complexes containing small RNAs in activating endosomal TLRs and cytoplasmic RNA sensors.

### 6.1. Endosomal TLR-Mediated Responses

Endosomal TLRs (TLR3, TLR7, and TLR8) recognize diverse viral RNA moieties and instigate host-protective responses. TLR3 recognises dsRNA, while both TLR7 and TLR8 sense ssRNAs. The inhibition of endosomal TLR signalling has been reported to ameliorate lupus in a mouse model [63]. Fcγ receptor-mediated IC endocytosis may expose Ro60 RNP autoantibody complexes to endosomal TLRs (Figure 3B). ICs containing Ro60 RNPs from apoptotic fibroblasts stimulate macrophages via TLR7, inducing the macrophages to secrete TNFα [50]. Neonatal lupus, characterised by heart conduction abnormalities and foetal congenital heart block (CHB), is caused by the transfer of maternal autoantibodies across the placenta [64]. Irrespective of underlying rheumatic conditions, over 85% of women who give birth to babies with CHB have anti-Ro/anti-La/SSB antibodies, although only 2% of babies born from mothers with anti-Ro/anti-La/SSB antibodies develop CHB [53,64,65]. This suggests that additional factors beyond autoantibodies are necessary for the development of neonatal lupus [2]. Clancy et al. demonstrated that fibrosis induction in CHB may begin with TLR signalling. Ro60, bound RNAs (pre-5S rRNA and hY3), and anti-Ro60 IgG from CHB stimulated macrophages to secrete TNFα in a manner dependent on FcγR and TLR7 [2].

SLE-associated nephritis is commonly marked by immunoglobulin deposition and leukocyte recruitment at the glomeruli. The administration of anti-TLR7 monoclonal antibodies [66] or the TLR7/TLR9 inhibitor IRS954 [63] diminishes autoantibody production and glomerulonephritis in lupus-prone mice, suggesting that RNA-stimulated TLR7 signalling promotes leukocyte recruitment and autoantibody production in SLE [66]. ICs containing Ro60 and associated small RNAs might engage with autoreactive B cells in a manner dependent on antigen-specific BCR, complement receptors, FcγRIIB, and TLR7 [67] (Figure 3C). TLR recognition is dependent on RNA sequence complementarity or uridine content, and Fc receptor engagement is required to bring the ICs into the endosomal compartment for TLR recognition.

### 6.2. Activation of Inflammasomes

Inflammasomes are a group of innate intracellular immune receptors that are activated in response to multiple damage-associated and pathogen-associated stimuli, including viral nucleic acids [68]. Inflammasome activation leads to the caspase-1-mediated processing and release of inflammatory cytokines IL-1β and IL-18 and to pro-inflammatory pyroptotic cell death. Potential inflammasome roles in autoimmune diseases pathogenesis have been reviewed extensively [69,70,71,72]. In RA, dysregulated fibroblast–macrophage crosstalk drives the release of pro-inflammatory cytokines, including TNFα, IL-1β, and IL-18. It is hypothesised that dysregulated pyroptotic cell death may contribute to driving this autoimmune disease [71]. SjD patients’ PBMCs exhibited increased expression of the NLR family pyrin domain containing 3 (NLRP3) inflammasome components ASC and caspase-1, as well as the NLRP3-induced cytokines IL-1β and IL-18, compared to that in healthy controls [73] (Figure 3B). Furthermore, immunofluorescent staining of SG-infiltrating macrophages from SjD patients showed evidence of increased pyroptotic activity by ASC speck formation compared to that in healthy controls. The increased gene expression of inflammasome components was then correlated with anti-Ro/SSA positivity and higher SG focal scores, suggestive of increased disease severity [74]. We speculate that pathogens which enter the body orally may activate innate immune pathways such as pyroptosis, leading to epithelial cell death [75]. For SjD pathogenesis, this may drive antigenic exposure, contributing to excessive autoimmune inflammation and SG dysfunction.

It remains unclear whether Ro60 RNPs can directly contribute to NLRP3 inflammasome activation and pyroptotic cell death. The NLRP3 inflammasome can be activated by cytosolic viral RNA [76], and it has been proposed that endogenous RNAs and retroelements may also be able to activate NLRP3 [77,78]. As Ro60 is known to bind endogenous retroelement transcripts [3], it is possible that Ro60 may stabilise or promote retroelement-specific NLRP3 activity in a manner similar to that reported for DEAD/H-box RNA-binding proteins [79,80]. Furthermore, NLRP3 is a key target of TLR7 signalling, and Ro60-associated RNAs may augment NLRP3 inflammasome expression indirectly through TLR7 [81]. Potential Ro60-NLRP3 interactions are worth further investigation.

### 6.3. Ro60 Antigen Presentation to T Cells

APCs can process and present peptides from internalised Ro60 to CD4^+^ T cells through the major histocompatibility complex (MHC)II pathway. APCs and other cells can process and present endogenous Ro60 to CD8^+^ T cells through MHCI. In normal conditions, self-tolerance mechanisms should prevent immune responses to autologous Ro60. Ro60 antigenic epitopes for T cells have been studied in various autoimmune conditions. In SjD, a dominant Ro60 epitope is the peptide sequence ELYKEKALSVETEKLLKYLEAV, between amino acids 211 and 232 within the RNA-binding TROVE domain [82]. In SLE, Ro60 epitopes were identified in amino acids 169–190 and also in the TROVE domain [82,83,84]. MHCII expression is predominantly a feature of APCs, and its expression in non-APCs can be defined as a pathogenic feature. The immunisation of mice with adjuvant-emulsified Ro60_316–335 led to the infiltration of T and B cells into the lacrimal gland and SGs, as well as induction of ectopic MHCII expression, consistent with a SjD-like process [85,86]. Characterising Ro60 epitopes may help to understand the immune response and develop targeted therapeutic approaches for autoimmune diseases where Ro60 is an autoantigen.

### 6.4. B-Cell Activation by Ro60 RNP

Self-reactive B cells can engage with the Ro60 RNP complex and its derivates (Figure 3C). These interactions can include those between Ro60-specific BCR and Ro60 RNP complexes, antigen-specific BCR and Ro60 ICs, and FcγRIIB and Ro60 ICs, as well as the co-engagement of FcγRIIB and antigen-specific BCR (Figure 3C). BCR and IC engagement can promote endosomal TLR signalling, potentially inducing an IFNα positive feedback loop and driving extrafollicular B-cell responses, germinal centre reactions, and autoantibody production [87] (Figure 3C). B cells express the inhibitory receptor FcγRIIB, which may provide negative feedback as a self-tolerance mechanism. FcγRIIB co-engagement with BCR-IC negatively regulates BCR-TLR7 endosomal response pathways [88]. Additional interactions between Ro60-RNP complexes and B cells can occur via complement receptors, which can reduce B-cell activation thresholds, whilst FcγRIIB engagement dampens B-cell activation and autoimmunity [89,90]. SLE patients are reported to exhibit reduced expression of FcγRIIB [90]. The dysregulation of BCR and FcγRIIB signalling promotes an autoinflammatory loop in an interleukin-1 receptor-associated kinase 4-dependent manner [91]. B-cell-intrinsic TLR7 signalling promotes anti-RNA antibody synthesis and lupus pathogenesis in lupus-prone mice [92]. It will be important to uncover whether and how Ro60 RNP complexes promote TLR7 signalling to drive pathogenesis in other autoimmune diseases.

## 7. Ro60 Autoantibodies in the Pathogenesis of Autoimmune Diseases

Are autoantibodies to Ro60 directly pathogenic, or are they simply a marker of other pathogenic mechanisms? A study of serial blood samples from US military personnel subsequently diagnosed with SLE revealed that anti-Ro60, anti-Ro52, and anti-La autoantibodies could be detected in the serum years before more SLE-specific autoantibodies targeting double-stranded DNA, Sm, and nuclear ribonucleoprotein antigens [93]. This observation is consistent with a model in which earlier autoantibody responses contribute to epitope spreading and the recognition of additional autoantigens as disease develops.

Potential pathogenic effects of autoantibodies have previously been reviewed extensively, particularly in the context of RA [94]. Specifically for anti-Ro60 autoantibodies, the pathogenic effects of these antibodies are closely linked to SLE and particularly to the pathogenesis of CHB [95,96]. Whilst the specific mechanism by which anti-Ro60 autoantibodies promote CHB remains unclear, one proposed mechanism involves the opsonisation of apoptotic foetal cardiomyocytes, driving the secretion of TNFα and tissue damage [97]. Alternatively, one theory suggests that anti-Ro60 antibodies bind to Ro60 antigen, which is redistributed to the surface during apoptotic cell death. Such binding is proposed to increase the expression of urokinase plasminogen activator receptor, which acts to inhibit efferocytosis. Thus, apoptotic cells are not cleared, leading to cardiac tissue fibrosis and eventual CHB [98]. For SjD, the pathogenic roles of anti-Ro60 autoantibodies are poorly understood despite the detection of these autoantibodies in both serum [99] and saliva [100]. Anti-Ro60 antibodies can appear years before a SjD diagnosis. In a retrospective analysis of biobank samples, Jannson et al. found that the median time between a positive test result for anti-Ro60 antibodies and the onset of SjD symptoms was 4 years, with a range of 1–18 years [101]. Seropositivity for SjD autoantibodies has been confirmed as early as 18–20 years before the onset of SjD [102], suggesting that the early detection of anti-Ro60 antibodies can serve as a better prognostic marker in patient care. However, it is possible that in patient SGs, autoantibody ICs may stimulate TLRs and B-cell antigen receptors, leading to chronic B-cell activation, dysregulated Type I IFN signalling, and feedforward autoimmune mechanisms [103]. Together, these pro-inflammatory immune responses may drive epithelial damage and SG dysfunction [75].

## 8. Ro60 Autoantibodies in the Diagnosis and Monitoring of Autoimmune Diseases

Ro60 is an extractable nuclear antigen, and serum Ro60 autoantibodies are common diagnostic markers for various autoimmune diseases, including SjD, SLE, SSc, inflammatory myositis, and neuropsychiatric SLE [60,99,104]. In keeping with the broader observation that serum autoantibodies foreshadow the development of autoimmune diseases [93,105], Ro60 autoantibodies can be detected years before the clinical onset of autoimmune diseases [106,107]. Anti-Ro60 autoantibodies may co-exist with autoantibodies targeting other RNPs, such as La/SSB, and the E3 ubiquitin ligase Ro52 (TRIM21). Not all diagnostic testing distinguishes autoantibodies targeting Ro60 from those targeting Ro52; these may be grouped under the term ‘SSA’ or ‘SSA/Ro’, with ‘Ro’ referring to a particular patient serum sample [45]. Blood is the most common biosample used in the diagnosis of autoimmune diseases; however, depending on the autoimmune condition, tears, saliva, and cerebrospinal fluid (CSF) may also be used [100,108]. Anti-Ro/SSA antibodies are not standalone markers; therefore, the detection of anti-SSA antibodies is accompanied by other diagnostic modalities [109,110].

### 8.1. Clinical Laboratory Techniques for Anti-Ro60 Detection and Measurement

Laboratory techniques used for the detection of anti-Ro (Ro60 and Ro52) autoantibodies include immunofluorescence (e.g., antinuclear antibody (ANA) staining with Hep-2 cells), immunohistochemistry, enzyme-linked immunosorbent assays (ELISAs) [111], fluorescence enzyme-linked immunoassays [112,113], chemiluminescence assays [113], double immunodiffusion [113], immunoprecipitation [113], immunoblotting, line/dot immunoassays [113,114,115], mass spectrometry [114], multiplex bead assays [113,116], and autoantigen microarrays. These methods present different strengths and weaknesses. For instance, immunohistochemistry and immunofluorescent staining-based methods (such as antinuclear antibody staining) may not provide sufficient information to identify specific target antigens. Line or dot immunoassays, ELISAs, autoantigen arrays, and multiplex bead arrays have the advantage that the antigen can be precisely defined. ELISAs and autoantigen arrays can be automated. Line immunoblot assays can simultaneously detect multiple autoantibody species and well-defined autoantigens, offering high specificity and low cross-reactivity; however, line immunoassays do not provide the same level of quantitative information as ELISA [115]. The UniCAP EliA SS-A/Ro, a fluorescence ELISA, can be configured to detect both anti-Ro60 and anti-Ro52 antibodies. Immunoblotting has high specificity in the detection of anti-Ro60 antibodies, which can be further validated by mass spectrometry analysis of excised bands [114]. Autoantigen microarrays and multiplexed assays can simultaneously detect multiple autoantibodies, offering a broader overview of disease status while requiring lower sample volumes than individual assays. However, cross-reactivity can result in false positive results [109]. Multiplex bead arrays can analyse reactivity to multiple antigens in small serum samples [116,117]. Whilst multiplex assays can simultaneously detect multiple antigens, often these results cannot be reproduced with conventional gold-standard assays. Additionally, if multiplex assays use antigen fragments or peptides, autoantibody reactivity binding may not accurately reflect the antigen–antibody interactions occurring with the natural antigen [118].

Using serum specimens from 184 patients with autoimmune diseases and 50 controls, Qin et al. evaluated the analytical and clinical performance of line immunoassays, multiplex bead-based flow fluorescent immunoassays, and magnetic bar code immunoassays for the detection of antinuclear antibodies, including those targeting Ro60 and Ro52; they found that the three assays showed good agreement [119]. A comparative analysis of anti-Ro antibody test results from six state-of-the-art assays used to test 181 patients across 21 centres highlighted potential pitfalls in detection accuracy and discrepancies in test results among the various methods, including immunoprecipitation, double immunodiffusion, immunoblotting, fluoro-immuno-enzymatic assays, line immunoassays, ELISAs, chemiluminescence assays, and multi-bead immunoassays [113]. Therefore, anti-Ro60 test results produced from multiplexed assays should be cross-validated with conventional assays specifically testing anti-Ro60 antibodies. The sensitivity, specificity, positive and negative predictive values, and diagnostic accuracy of these diagnostic approaches may vary, and combinations of methods are recommended in clinical settings [110]. Diagnostic accuracy is vital for autoantibody detection, thus requiring the consistent refinement of validation assays.

### 8.2. Use of Ro60 Serology in Diagnosis and Monitoring

Ro/SSA autoantibody testing is key to the diagnosis and monitoring of multiple systemic autoimmune rheumatic diseases, including SLE, RA, SjD, SSc, and inflammatory myositis [99]. A retrospective analysis by Robbins et al. demonstrated that of 13032 ANA-positive individuals, 399 were positive for Ro60 and/or Ro52 antibodies. SjD was the most common diagnosis for Ro52^+^/Ro60^+^ patients, while SLE was most frequent in the Ro52^−^/Ro60^+^ group. However, in Ro52^+^/Ro60^−^ patients, the most frequent diagnoses were inflammatory myositis and inflammatory arthritis [99]. Menéndez et al. detected a 44% prevalence of anti-Ro60 and/or anti-Ro52 antibodies in SLE (62 of 141). Of the 62 seropositive individuals, 37 (26.2%) were seropositive for both anti-Ro52 and anti-Ro60, 23 (16%) were seropositive for anti-Ro60, and 2 (1.4%) were seropositive for anti-Ro52. Anti-Ro reactivities can also show strong corelation with clinical and immunological manifestations of SLE [104]. Anti-Ro52, anti-Ro60, and anti-La antibody test results may have distinct associations with SjD severity. Single anti-Ro60^+^ test results are associated with the least severe form of SjD, whilst single anti-Ro52 test results and anti-Ro60 and anti-Ro52 double-positive test results are associated with increased primary SjD severity. Triple-positive (anti-Ro52^+^/Ro60^+^/La^+^) test results were associated with the most severe SjD clinical picture in one study [120], while another study reported that isolated anti-Ro52^+^ results were associated with increased SjD severity [114]. In a cohort of 508 patients, individuals with combined reactivity for anti-Ro52/anti-Ro60 and anti-La antibodies had an increased risk of lymphomas, while anti-Ro52 single-positive test results were associated with inflammatory myositis, RA, and SjD [115]. Ro52^−^/Ro60^+^ test results were associated with oral ulcers [115]. Anti-Ro antibody test results can be used to monitor disease progression and therapeutic outcomes; in patients with RA, positive anti-Ro test results (Ro60 or Ro52 not specified) were associated with treatment resistance to infliximab, but not with tocilizumab and/or abatacept [121]. SjD can co-occur with other autoimmune diseases, including RA, SSc, SLE, and myositis. Anti-Ro60 antibody test results can have differential diagnostic implications for overlapping syndromes.

The classification of autoimmune patients into different categories can be achieved through approaches like counter immune electrophoresis followed by line immunoblotting to stratify patients into anti-Ro60^low^ and anti-Ro60^high^ groups. Ro60^low^ patients’ anti-Ro60 antibodies have fewer somatic mutations with restricted heavy variable gene use [110]. The diagnostic and prognostic significance of anti-Ro60 autoantibodies is not uniformly established, and comparative measures of sensitivity and specificity are lacking. Anti-Ro60, in combination with anti-RLP0, anti-RLP1, and anti-RLP2 autoantibodies in the CSF, may serve as diagnostic markers for neuropsychiatric SLE [109]. In SjD, the combined reactivity of anti-Ro52, anti-Ro60, and anti-La is associated with increased lymphoma risk [115]. Overall, the combinations of autoantibodies and their levels provide valuable information for the classification of disease, estimation of disease severity, and prediction of associated complications. It is also imperative to understand whether anti-Ro60 antibody levels in autoimmune patients vary with time and disease severity. Derksen et al. monitored anti-Ro60 antibodies over a period of 80 months spanning two pregnancies in an SLE patient and found that anti-Ro60 antibody levels fluctuated over time in a manner unrelated to disease severity index, anti-dsDNA antibody level, or immunosuppressive therapies [111]. Conversely, Lindop et al. reported that in the case of SjD, anti-Ro60 antibody levels remained stable from months to years [106].

Understanding autoantibody dynamics may be useful for monitoring and devising new therapies for autoimmune diseases. RA patients with a positive anti-SSA diagnosis are more likely to develop anti-drug antibodies, limiting the efficacy of the TNFα antagonists infliximab [121] and adalimumab [122]. Whilst an ‘anti-SSA-positive’ diagnosis could mean Ro52^+^/Ro60^−^, Ro52^−^/Ro60^+^, or Ro52^+^/Ro60^+^, the differential diagnosis of Ro60 and Ro52 antibodies has diagnostic and prognostic utility [99,120]. Ro52^−^/Ro60^+^ patients were more likely to have SLE, whilst Ro52^+^/Ro60^+^ patients were more likely to have SjD; Ro52^+^/Ro60^−^ patients were more likely to have other inflammatory diseases [99]. Furthermore, in SjD, a differential and combinatorial positive diagnosis of anti-Ro60, anti-Ro52, and anti-La/SSB antibodies was associated with different clinical features, suggesting the separate assessment of anti-Ro52 and anti-Ro60 antibodies for efficient patient stratification to allow for the development of tailored treatment plans [120]. To overcome all the potential confusion, it may be advisable to discontinue the use of the SSA acronym in favour of specifically designating Ro60 and Ro52.

## 9. Commensals, Ro60 Cross-Reactivity, and Autoimmunity

Commensal microorganisms—our microbiome—reside in and on various part of the body, including the gastrointestinal tract, skin, oral cavity, and other mucosal surfaces. The diversity and composition of the microbiome vary among different individuals and across different sites within the body of same individual. The human gastrointestinal tract can contain 10^13^–10^14^ microorganisms, equalling or exceeding the number of human cells in the body, and with up to ~100 times more genetic information than the human genome [123]. Dysbiosis, i.e., changes in microbiome composition or the overgrowth of certain microbes, has been associated with various pathophysiological conditions, including autoimmune diseases [124].

Ro60 is evolutionarily conserved, and its orthologues are present in commensal bacteria. Greiling et al. identified Ro60 orthologues in subsets of microbes that colonise the human oral mucosa, skin, and gut [125]. Antibodies from Ro60-seropositive SLE patient sera immunoprecipitated bacterial Ro60 RNPs from the normal microbiome constituent species *Propionobacterium propionicum* (*Pp*, oral and skin microbiomes) and *Bacteroides thetaiotaomicron* (*Bt*, gut microbiome). The immunoprecipitated RNPs included bacterial YRNAs. Peptides from *Pp* and *Bt* Ro60 also stimulated Ro60-reactive T-cell clones from SLE patients. Germ-free mice developed T- and B-cell responses cross-reactive to human Ro60 following colonisation with *Bt* [125]. In another study, Szymula et al. reported that peptides derived from the TROVE and vWFA domains of bacterial Ro60 orthologues could stimulate T-cell hybridomas specific for human Ro60 epitopes [126]. The bacterium *Capnocytophaga ochracea*, which is part of the normal oral microbiome, was identified as a potential source of cross-reactive Ro60 peptides [126]. These reports suggest that Ro60-autoreactive cells may respond to normal constituents of the human microbial flora; importantly, patients with Ro60 autoimmunity did not exhibit readily identifiable changes in microbiome makeup compared to healthy volunteers [125]. The antigenic cross-reactivity of orthologues and dysbiosis should be considered potential causal mechanisms that may disrupt self-tolerance and, if left unchecked, lead to autoimmunity.

## 10. Conclusions and Future Perspectives

Ro60 is a highly conserved protein which binds and controls the cellular distribution of small RNAs, in particular YRNAs and rRNAs. Importantly, Ro60 provides cellular surveillance by binding retroelements such as Alu transcripts, which may otherwise drive inflammation, excessive RNA editing, mutagenesis, and possible malignancies. When bound to small RNAs, Ro60 RNPs may also contribute to innate immune processes such as NLRP3 inflammasome activation. However, our understanding of these possible interactions is sparse and warrants further investigation. Ro60 as an autoantigen is well established in multiple autoimmune diseases, including SjD, SLE, RA, and SSc, in which anti-Ro60 autoantibodies are commonly detected. However, the mechanisms by which Ro60 is targeted by the immune response remain unclear. In this review, we outline various potential mechanisms, including the secretion of Ro60 RNP complexes within EVs, exposure during immunogenic forms of cell death, TLR-mediated stimulation, and possible cross-reactivity from commensal microorganisms. Further studies should explore whether such processes truly drive anti-Ro60 autoantibody production. In vitro studies could investigate possible contributions of Ro60 RNPs to forms of pro-inflammatory lytic cell death and assess how EVs containing Ro60 RNPs are taken up and enable a breach in self-tolerance. Together, these studies will contribute to our understanding of how and why Ro60 becomes an autoantigen.

## Figures and Tables

**Figure 1 ijms-25-07705-f001:**
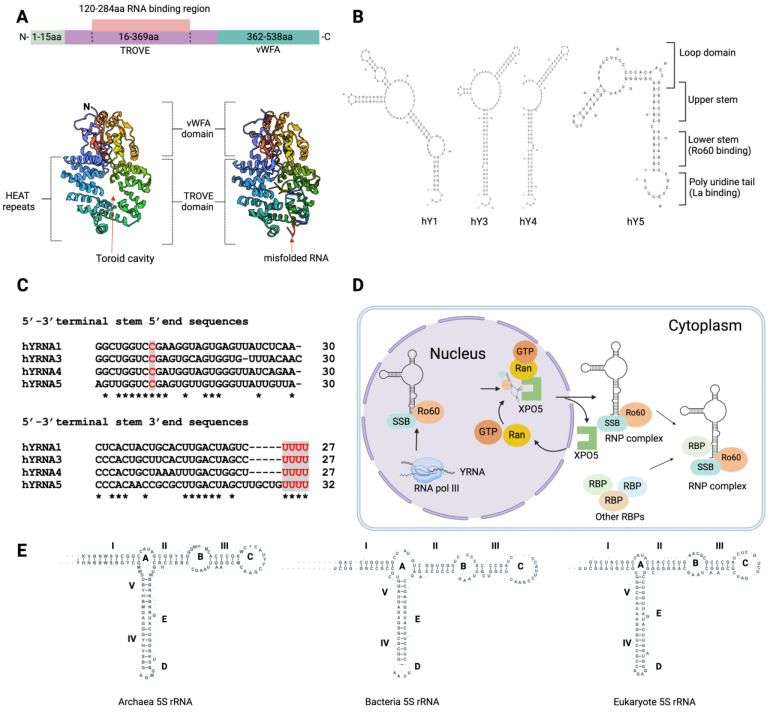
Ro60/SSA2 domain structure and small RNA structural and functional surveillance for cellular distribution. (**A**) Ro60 comprises two major domains, vWFA and TROVE (1YVR and 1YVP). The TROVE domain is composed of toroid HEAT repeats. (**B**) Schematic representation of human Ro60 and other autoantigenic RBPs bound to small RNAs (such as YRNAs) and their nuclear–cytoplasmic transportation via XPO5/Ran GTPase [21]. (**C**) UNAfold predicted secondary structure of human YRNAs (hY1, hY3, hY4, and hY5). * designates conserved residues. (**D**) YRNA terminal stem 5′ and 3′ end sequences. Ro60 binds to the 5′-3′ stem-loop bulge, while La binds to the 3′ terminal uridine oligos. (**E**) Secondary structure of 5S rRNAs from bacteria, archaea, and eukaryotes (structures adapted from the 5S rRNA database: http://combio.pl/rrna/alignment/, Accessed on 16 May 2024). A designates the three-helix junction hinge region; C and D represent hairpin loops; B and D designate two internal loops; I–V designate helical stems.

**Figure 2 ijms-25-07705-f002:**
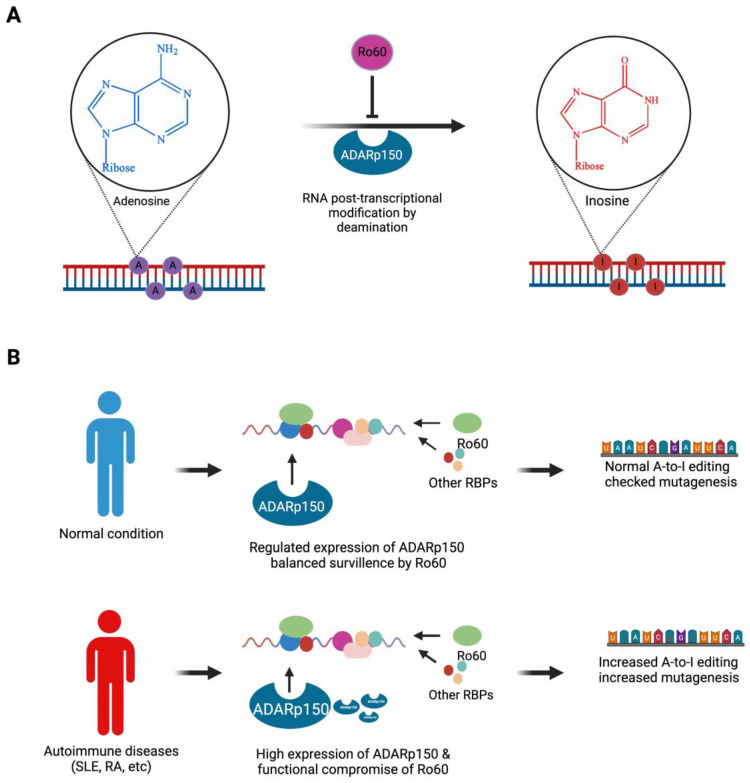
Ro60’s role in ADAR-mediated RNA editing. (**A**) Post-transcriptional RNA editing: adenosine-to-inosine editing is catalysed by ADARs (ADAR1, 2, and 3) through hydrolytic deamination, in which ADARp150 is a sentinel of autoimmunity. (**B**) Ro60 serves as a regulator of RNA editing by mediating the structural surveillance of RNA moieties. In autoimmune diseases, due to the elevated expression of ADAR1p150 enzyme and retroelements L1 and Alu, structural surveillance by Ro60 is compromised, and excessive RNA editing occurs. Excessive RNA editing by ADARp150 in autoimmune diseases can be mutagenic, which may increase the risk of malignancy in patients with systemic autoimmune rheumatic disease.

**Figure 3 ijms-25-07705-f003:**
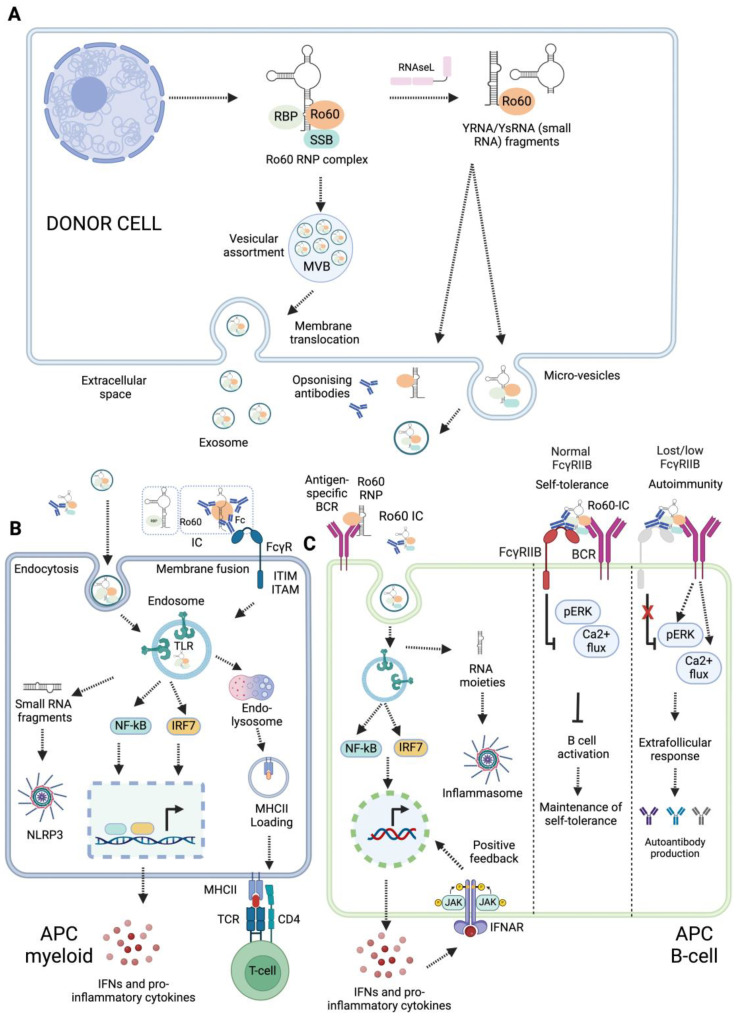
Extracellular presentation and innate immune sensing of Ro60 RNP complex. (**A**) Donor cells can liberate Ro60 RNP complex in extracellular space via three basic mechanisms, small RNA-mediated translocation, extracellular vesicle-mediated translocation, and apoptotic body-mediated translocation. (**B**) The presentation of Ro60 RNP complex to myeloid APCs may promote endosomal TLR signalling and the production of pro-inflammatory cytokines and chemokines. (**C**) Ro60 RNP complex presentation to antigen-specific BCR and FcγRIIB, promoting endosomal TLR7 signalling, inflammation, and production of autoantibodies. IFNs can mount positive feedback on the transcription of IFN-stimulated genes. FcγRIIB regulates the activation of B cells. The loss or reduced expression of FcγRIIB is associated with an extrafollicular response and the production of autoantibodies.

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
