# Peer review of "Ro60—Roles in RNA Processing, Inflammation, and Rheumatic Autoimmune Diseases"

_ijms, 2024, doi:10.3390/ijms25147705_

Round 1

Reviewer 1 Report

Comments and Suggestions for Authors

Mahla and colleagues have written an engaging review on the role of anti-Ro60 antibodies. They have effectively described their biological functions and their role in inflammatory processes.

I would suggest including a section on laboratory techniques used to identify these antibodies, highlighting their advantages and disadvantages.

The section on monitoring rheumatic diseases is rather limited, and their role in different autoimmune diseases is not described in detail. However, I understand that this section may be slightly off-topic from the main focus of the review. If possible, I would add their pathogenetic role at least in Sjogren's syndrome and mention their involvement in inflammatory myositis.

Overall, this is an excellent review.

Author Response

Comment 1:  Mahla and colleagues have written an engaging review on the role of anti-Ro60 antibodies. They have effectively described their biological functions and their role in inflammatory processes.

Response 1: We thank the reviewer for their careful assessment of the manuscript.

Comment 2: I would suggest including a section on laboratory techniques used to identify these antibodies, highlighting their advantages and disadvantages.

Response 2: Thank you for this suggestion. Please see the new section 8.1 on laboratory techniques (lines 421-463).

Comment 3: The section on monitoring rheumatic diseases is rather limited, and their role in different autoimmune diseases is not described in detail. However, I understand that this section may be slightly off-topic from the main focus of the review. If possible, I would add their pathogenetic role at least in Sjogren's syndrome and mention their involvement in inflammatory myositis.

Response 3: Thank you for this suggestion. We have discussed potential pathogenic roles for anti-Ro60 in lines 378-405 and 409-412. Inflammatory myositis is mentioned in lines 409 and 464-486. 

Comment 4: Overall, this is an excellent review.

Response 4: thank you!

Reviewer 2 Report

Comments and Suggestions for Authors

This review outlines different potential mechanisms included in secretion of Ro60 and it is role in provocation of various autoimmune diseases. the review is comprehensive and identified the gap of potential mechanisms included in autoimmune disease through extracellular secretion of Ro 60. The Most of cited references within 10 years. the conclusion is supported by listed citations and figures  appropriately show the data, clear,  and easy to understand. 

Author Response

Comment 1: This review outlines different potential mechanisms included in secretion of Ro60 and it is role in provocation of various autoimmune diseases. the review is comprehensive and identified the gap of potential mechanisms included in autoimmune disease through extracellular secretion of Ro 60. The Most of cited references within 10 years. the conclusion is supported by listed citations and figures  appropriately show the data, clear,  and easy to understand. 

Response 1: We thank the reviewer for these encouraging remarks.

Reviewer 3 Report

Comments and Suggestions for Authors

The article comprehensively reviews the Ro60 autoantigen, detailing its roles in RNA processing, inflammation, and its involvement in rheumatic autoimmune diseases. The depth of the review, combined with the breadth of topics covered, makes it a suitable candidate for publication. Including various figures and detailed explanations of Ro60's interactions with small RNAs and its implications in autoimmune diseases adds to its scientific merit. In the abstract, please define abbreviations, such as YRNA, when they appear first. This article has been published in PrePrints: Mahla, R. S.; Jones, E. L.; Dustin, L. B. Ro60 - Roles in RNA Processing, Inflammation, and Rheumatic Autoimmune Diseases. Preprints 2024, 2024060929. https://doi.org/10.20944/preprints202406.0929.v1. I am not sure if disclosure of this is required.

The article extensively covers the structural and functional aspects of Ro60, including its role in RNA metabolism, immune response, and involvement in autoimmune diseases like Sjögren’s syndrome and systemic lupus erythematosus (SLE). It provides detailed mechanistic insights into how Ro60 interacts with various RNAs, the pathways it influences, and how these interactions can lead to autoimmune responses. The figures in the article help us understand the complex interactions and structural components of Ro60, making the content more accessible to readers.

The review is well-supported by numerous references, ensuring the information is backed by existing literature. However, I recommend the authors consider adding the following citations and a brief description:

o   Scofield, R. H. (2004). Autoantibodies as predictors of disease. The Lancet, 363(9420), 1544-1546. This reference can provide additional context on the role of autoantibodies in predicting autoimmune diseases, supporting the discussion of anti-Ro60 autoantibodies as diagnostic markers.

o   Keene, J. D. (2001). RNA regulons: coordination of post-transcriptional events. Nature Reviews Genetics, 2(7), 529-537. This paper discusses the coordination of post-transcriptional events, which can provide a deeper understanding of RNA metabolism and its relevance to Ro60’s function.

o   Costa, Y., & Cooke, H. J. (2007). Dissecting the mammalian spermatogenesis process using RNA interference. Reproduction, 134(5), 707-719. This reference can provide insights into RNA quality control mechanisms that might be relevant to the role of Ro60.

o   Glisovic, T., Bachorik, J. L., Yong, J., & Dreyfuss, G. (2008). RNA-binding proteins and post-transcriptional gene regulation. FEBS Letters, 582(14), 1977-1986. This article provides a comprehensive overview of RNA-binding proteins and their roles, complementing the discussion on Ro60’s structure and function.

o   Barrat, F. J., Meeker, T., Chan, J. H., Guiducci, C., & Coffman, R. L. (2007). Treatment of lupus-prone mice with a dual inhibitor of TLR7 and TLR9 leads to reduction of autoantibody production and amelioration of disease symptoms. European Journal of Immunology, 37(12), 3582-3586. This reference can enhance the discussion on the immune response and the potential therapeutic implications of targeting TLR pathways in diseases involving Ro60.

o   Arbuckle, M. R., McClain, M. T., Rubertone, M. V., Scofield, R. H., Dennis, G. J., James, J. A., & Harley, J. B. (2003). Development of autoantibodies before the clinical onset of systemic lupus erythematosus. New England Journal of Medicine, 349(16), 1526-1533. This study can support the discussion on the clinical implications of anti-Ro60 antibodies and their presence before the onset of disease.

o   Kowalski, M. P., & Krude, T. (2015). Functional roles of non-coding Y RNAs. International Journal of Biochemistry & Cell Biology, 66, 20-29. This reference can provide additional information on the functional roles of YRNAs, complementing the sections discussing Ro60’s interaction with YRNAs.

Some sections of the article repeat information, particularly regarding the interaction between Ro60 and YRNAs. Consolidating these sections could improve the flow and readability. The review could benefit from a more explicit emphasis on novel contributions and gaps in the current understanding of Ro60. Highlighting these areas could strengthen the article’s impact.

The article touches on the diagnostic and therapeutic implications of anti-Ro60 autoantibodies but could benefit from a more in-depth discussion. Exploring how these findings could translate into clinical practice would add significant value.

Author Response

Comment 1: The article comprehensively reviews the Ro60 autoantigen, detailing its roles in RNA processing, inflammation, and its involvement in rheumatic autoimmune diseases. The depth of the review, combined with the breadth of topics covered, makes it a suitable candidate for publication. Including various figures and detailed explanations of Ro60's interactions with small RNAs and its implications in autoimmune diseases adds to its scientific merit. In the abstract, please define abbreviations, such as YRNA, when they appear first. 

Response 1: We thank the reviewer for their careful assessment of the manuscript. We have double checked that abbreviations are defined in the Abstract; the origin of the "Y" in "YRNA" is from the word cYtoplasm and distinguishes these small RNAs from other small RNAs found only in the nUcleus (URNAs).

Comment 2: This article has been published in PrePrints: Mahla, R. S.; Jones, E. L.; Dustin, L. B. Ro60 - Roles in RNA Processing, Inflammation, and Rheumatic Autoimmune Diseases. Preprints 2024, 2024060929. https://doi.org/10.20944/preprints202406.0929.v1. I am not sure if disclosure of this is required.

Response 2: The paper was uploaded to PrePrints as part of the submission process and is encouraged by the journal. 

Comment 3: The article extensively covers the structural and functional aspects of Ro60, including its role in RNA metabolism, immune response, and involvement in autoimmune diseases like Sjögren’s syndrome and systemic lupus erythematosus (SLE). It provides detailed mechanistic insights into how Ro60 interacts with various RNAs, the pathways it influences, and how these interactions can lead to autoimmune responses. The figures in the article help us understand the complex interactions and structural components of Ro60, making the content more accessible to readers.

Response 3: We appreciate the reviewer's careful reading and encouraging words.

Comment 4: The review is well-supported by numerous references, ensuring the information is backed by existing literature. However, I recommend the authors consider adding the following citations and a brief description:

o   Scofield, R. H. (2004). Autoantibodies as predictors of disease. The Lancet, 363(9420), 1544-1546. This reference can provide additional context on the role of autoantibodies in predicting autoimmune diseases, supporting the discussion of anti-Ro60 autoantibodies as diagnostic markers.

o   Keene, J. D. (2001). RNA regulons: coordination of post-transcriptional events. Nature Reviews Genetics, 2(7), 529-537. This paper discusses the coordination of post-transcriptional events, which can provide a deeper understanding of RNA metabolism and its relevance to Ro60’s function.

o   Costa, Y., & Cooke, H. J. (2007). Dissecting the mammalian spermatogenesis process using RNA interference. Reproduction, 134(5), 707-719. This reference can provide insights into RNA quality control mechanisms that might be relevant to the role of Ro60.

o   Glisovic, T., Bachorik, J. L., Yong, J., & Dreyfuss, G. (2008). RNA-binding proteins and post-transcriptional gene regulation. FEBS Letters, 582(14), 1977-1986. This article provides a comprehensive overview of RNA-binding proteins and their roles, complementing the discussion on Ro60’s structure and function.

o   Barrat, F. J., Meeker, T., Chan, J. H., Guiducci, C., & Coffman, R. L. (2007). Treatment of lupus-prone mice with a dual inhibitor of TLR7 and TLR9 leads to reduction of autoantibody production and amelioration of disease symptoms. European Journal of Immunology, 37(12), 3582-3586. This reference can enhance the discussion on the immune response and the potential therapeutic implications of targeting TLR pathways in diseases involving Ro60.

o   Arbuckle, M. R., McClain, M. T., Rubertone, M. V., Scofield, R. H., Dennis, G. J., James, J. A., & Harley, J. B. (2003). Development of autoantibodies before the clinical onset of systemic lupus erythematosus. New England Journal of Medicine, 349(16), 1526-1533. This study can support the discussion on the clinical implications of anti-Ro60 antibodies and their presence before the onset of disease.

o   Kowalski, M. P., & Krude, T. (2015). Functional roles of non-coding Y RNAs. International Journal of Biochemistry & Cell Biology, 66, 20-29. This reference can provide additional information on the functional roles of YRNAs, complementing the sections discussing Ro60’s interaction with YRNAs.

Response 4: These citations, and additional citations, have been added where appropriate in the manuscript (e.g. lines 36-39, 290-291, 305-307, 409-412). We were unable to locate the reference by Costa, Y., & Cooke, H. J. (2007). 

Comment 5: Some sections of the article repeat information, particularly regarding the interaction between Ro60 and YRNAs. Consolidating these sections could improve the flow and readability. The review could benefit from a more explicit emphasis on novel contributions and gaps in the current understanding of Ro60. Highlighting these areas could strengthen the article’s impact.

Response 5: We have endeavoured to consolidate the discussion where possible. It has been challenging to perform a thorough search for novel contributions about Ro60 in the very short time permitted for revising this manuscript, but we have done our best to discuss the potential roles of Ro60 autoantibodies in pathogenesis.

Comment 6: The article touches on the diagnostic and therapeutic implications of anti-Ro60 autoantibodies but could benefit from a more in-depth discussion. Exploring how these findings could translate into clinical practice would add significant value.

Response 6: Thank you for this comment. We have extended our discussions of diagnostic methods (see lines 421-463) and the therapeutic implications of Ro60 autoantibodies (see lines 464-486 along with lines 487-519).